# A Review of Neurotransmitters Sensing Methods for Neuro-Engineering Research

**Shimwe Dominique Niyonambaza [1,2], Praveen Kumar [1], Paul Xing [3], Jessy Mathault [1], Paul De Koninck [1,4], Elodie Boisselier [2,5], Mounir Boukadoum [6] and Amine Miled [1,*]**

[1] LABioTRON Bioeng. Research Laboratory, ECE Dept. Université Laval, Québec City, QC G1V 0A6, Canada; shimwe-dominique.niyonambaza.1@ulaval.ca (S.D.N.); praveenkumar.kaveri.1@ulaval.ca (P.K.); jessy.mathault.1@ulaval.ca (J.M.); Paul.DeKoninck@neurosciences.ulaval.ca (P.D.K.)

[2] CUO-Recherche, Hôpital du Saint-Sacrement, Québec City, QC G3K 1A3, Canada; Elodie.Boisselier@fmed.ulaval.ca

[3] Neurosciences Dept. University of Montreal, Montreal, QC H3C 3J7, Canada; paul.xing@umontreal.ca

[4] CERVO Brain Research Center, Québec City, QC G1J 2G3, Canada

[5] Ophthalmology Department, Faculty of Medicine, Université Laval, Québec City, QC G1V 0A6, Canada

[6] CoFaMic, Université du Québec à Montréal, Montreal, QC H2L 2C4, Canada; boukadoum.mounir@uqam.ca

[*] Correspondence: amine.miled@gel.ulaval.ca; Tel.: +1-(418)-656-2131 (ext. 8966)

**Abstract:** Neurotransmitters as electrochemical signaling molecules are essential for proper brain function and their dysfunction is involved in several mental disorders. Therefore, the accurate detection and monitoring of these substances are crucial in brain studies. Neurotransmitters are present in the nervous system at very low concentrations, and they mixed with many other biochemical molecules and minerals, thus making their selective detection and measurement difficult. Although numerous techniques to do so have been proposed in the literature, neurotransmitter monitoring in the brain is still a challenge and the subject of ongoing research. This article reviews the current advances and trends in neurotransmitters detection techniques, including in vivo sampling and imaging techniques, electrochemical and nano-object sensing techniques for in vitro and in vivo detection, as well as spectrometric, analytical and derivatization-based methods mainly used for in vitro research. The document analyzes the strengths and weaknesses of each method, with the aim to offer selection guidelines for neuro-engineering research.

**Keywords:** neurotransmitters; sensing methods; nuclear medicine tomographic imaging; optical sensing techniques; electrochemistry; high performance liquid chromatography; microdialysis

## 1. Introduction

Most human body functions such as perception, motor control and cognitive behaviour are linked to the nervous system [1–14]. They are controlled by neural cell networks that generate and propagate electrochemical signals, in great part through the release and diffusion across synaptic junctions of neurotransmitters that mediate the exchange of those signals between the neural cells. A synapse involves a presynaptic neuron and a postsynaptic neuron that release and bind neurotransmitters, respectively.

As neurotransmitters are essential components in the nervous system, and in order to understand how the brain works and identify neurological diseases, multiple approaches have been proposed for neurotransmitter detection and monitoring. Depending on the used technology, the regularly used detection methods can be classified into five categories: (1) Nuclear medicine tomographic imaging, including positron emission tomography and single-photon emission computed

tomography; (2) Optical sensing, including surface-enhanced Raman spectroscopy, fluorescence, chemiluminescence, optical fiber based biosensors and colorimetry; (3) Electrochemical detection, including voltammetry and amperometry; (4) Analytical chemistry techniques, including high performance liquid chromatography; and (5) Microdialysis. In practice, two or more techniques may be combined for better detection accuracy. However, the detection of neurotransmitter is still challenging due to their sparsity in the brain and their mixture with other molecules. Thus, for accurate detection and monitoring, a neurotransmitter has to be localized and identified among other biomolecules and other neurotransmitters, as they can have similar behaviours.

In order to qualify as a neurotransmitter, a molecule must meet the following criteria, according to Kandel et al. [15]:

- It must be synthesized and released by the same neuron and stocked at the presynaptic terminal;
- Its release induces a specific behavior on the postsynaptic neuron;
- Exogenous administration must generate the same effect;
- Its induced action on the postsynaptic cell can be stopped by a specific mechanism.

The number of known neurotransmitters molecules exceeds 100 [16] and, based on their chemical structure, they are generally classified as amino acids, monoamines, neuropeptides, purines and gasotransmitters among others. Figure 1 shows the main locations of the brain where representative neurotransmitters are found in the highest concentrations.

Although all neurotransmitters are essential to the good functioning of the central and peripheral nervous system, only a small group is involved in the majority of known neuropathies. As a result, the existing detection techniques are usually optimized for specific neurotransmitters, of which those of the monoamine and amino-acid groups and others such as acetylcholine. A summary description of those neurotransmitters follows:

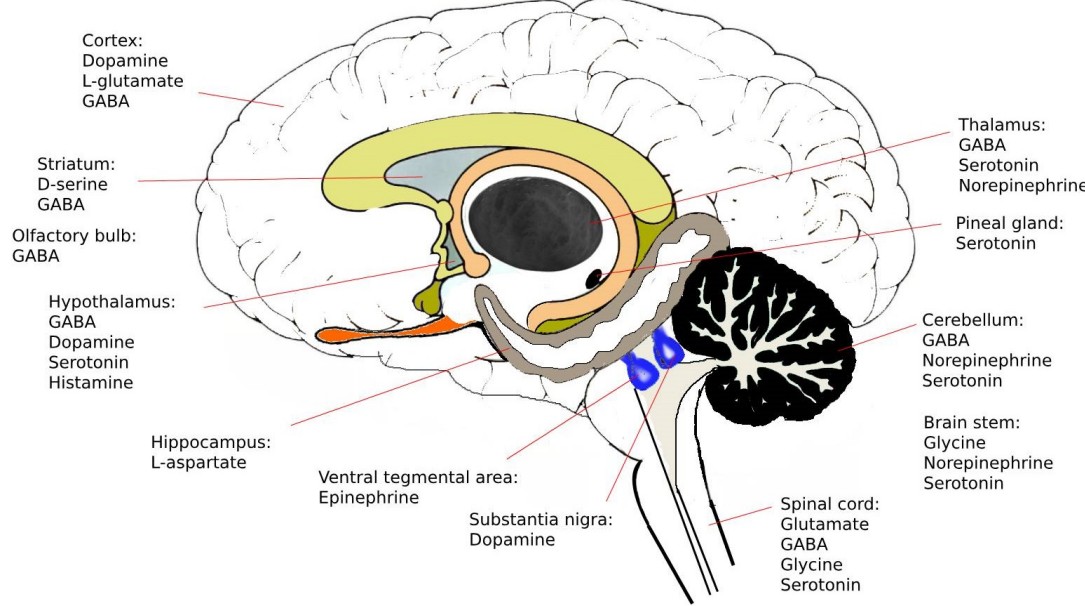

**Figure 1.** Regions of the brain where neurotransmitters are mainly produced.

## 1.1. Amino-Acids Neurotransmitters

### 1.1.1. Glutamate

Glutamate is the major excitatory neurotransmitter in the central nervous system [15]. It is present in 90% of the excitatory synapses [17] and plays a key role in the plasticity of the nervous system (defined as the ability of neuronal circuits to change their connectivity) [15,18–20]. Recent studies

have shown that glutamate also plays a role in glia-neuron signalling, along with two other neurotransmitters: D-serine and glycine [21,22].

Glutamate (and gamma-aminobutyric acid (GABA), defined further below) dysfunctions may be involved in brain disorders such as epilepsy [23–25]. Indeed, seizure was observed in mice lacking the glutamate transporter GLT-1 [24]. Besides that, the use of microdialysis and electroencephalography to measure extracellular glutamate in patients with epilepsy has shown a hippocampal increase of this neurotransmitter [25]. Recently, a glutamate dysfunction model of schizophrenia has emerged in addition to the widely known hypothesis involving dopamine. Schizophrenia is characterized by the disruption of sensory processing, leading to hallucination, paranoia, cognitive deficits in attention, short-term memory and movement abnormality [26].

### 1.1.2. L-Aspartate

L-aspartate is a conjugate base of the aspartic acid, whose role as a neurotransmitter in the central nervous system has been subject to controversy [27,28]. Some researchers have proposed that L-aspartate may be a neurotransmitter in the visual cortex and in the cerebellum [29–32], and others have shown that, like L-glutamate, L-aspartate may be an excitatory neurotransmitter and a neuropeptide-like modulator in the hippocampus [27,33].

The synaptic terminations that contain aspartate vesicles are co-localized with neurons containing glutamate and GABA vesicles. Thus, it seems that L-aspartate can play a role in both excitatory and inhibitory pathways. Synaptic termination with only aspartate vesicles has not been discovered in the hippocampus. A role for aspartate as co-neurotransmitter with glutamate was previously reported in the rat corticostriatal neurons and its release may be modulated by dopamine [34–37].

### 1.1.3. Gamma-Aminobutyric Acid (GABA)

GABA is the major inhibitory neurotransmitter in the central nervous system [15]. It is synthesized from glutamic acid [15].

Studies have shown that GABA is excitatory in early development [38]. Indeed, it induces a depolarization instead of hyperpolarization in various area of the nervous system, including the neocortex, hippocampus, hypothalamus, cerebellum and spinal cord. This is due to the higher chloride concentration in neurons during the early development of the human body, leading to an outward instead of inward chloride flux [38]. Therefore, the role of GABA during the first years of youth is not the same as in adults. The change of expression of two categories of chloride transporters, the sodium potassium chloride co-transporters and the potassium chloride co-transporters mediate the change of GABA action from excitatory to inhibitory [38]. Ben-Ari proposed that this GABA action switch promotes synapse formation [38]. He proposed the following three-stage mechanism [38]:

- GABA initially enables an excitatory effect on neurons due to the expression of transporters.
- Glutamategic synapses are formed after the GABAergic neurons.
- GABA and glutamate excitatory action enables oscillation of intracellular calcium that modulates synapse formation.

Poo et al. also showed that the excitatory action of GABA plays a major role in the morphological maturation of cortical neurons [39].

### 1.1.4. Glycine

Glycine is the major inhibitory neurotransmitter in the spinal cord, with a higher concentration at the ventral horn [15,40–42]. Glycine also acts as a neurotransmitter in the brainstem and medulla, and is also a required co-agonist with glutamate for N-methyl-d-aspartate (NMDA) receptors [43]. It is involved in voluntary motor control and sensory processing, and also in auditory, cardiovascular and respiratory functions [41,42]. There are glycine receptors in the bipolar, ganglion and amacrine

cells of the retina, with different subunit expressions [44]. As with GABA, Glycine acts as an excitatory neurotransmitter in early development, and may thus play a role in neuronal differentiation, proliferation, and connectivity [41,44].

The mutations in glycine receptors seem to be involved in motor control disorders like hypertonia and hyperekplexia [40,42]. Glycine transmission may also be involved in alcohol intoxication [42].

Studies have shown that GABA and glycine are co-released in certain neurons in the cerebellum, spinal cord and superior olive, but the interaction between the two neurotransmitters is not well understood [41]. Both glutamate and glycine sites must be occupied to lead to channel opening, with glutamate being released by the presynaptic neurons and glycine by glial cells [42].

### 1.1.5. D-Serine

D-serine is a molecule released by glial cells to act as a neurotransmitter or neuromodulator [45]. The concept of a functional role of D-configuration amino acids in higher organisms like mammals is relatively new, since they were thought to be only used in less complex living organism forms such as bacteria or insects [46,47]. It is present in various parts of the brain and immunohistochemical studies have shown its presence in the rostral cerebral cortex, hippocampus, anterior olfactory nuclei, olfactory tubercule, corpus striatum, and amygdala [46–49]. More precisely, D-serine is localized in the protoplasmic astrocytes of the grey matter that ensheath synapses [46,47].

Studies have shown that D-serine is involved in the glutamatergic hypothesis of schizophrenia [47]. Studies with schizophrenic patients showed that D-serine-based treatment seems to be effective for positive and negative symptoms [50,51].

### *1.2. Monoamine Neurotransmitters*

Monoamines contain one amino group connected to an aromatic ring by a two-carbon chain. They are classified into three groups: catecholamines, indolamines and imidazoleamines. The three known catecholamines (dopamine, norepinephrine and epinephrine) are all derived from the same precursor, L-DOPA, which is itself derived from the amino acid tyrosine [15]. Indolamines such as serotonin, melatonin and tryptamine all contain an indole compound ring. Imidazoleamines such as histamine contain an imidazole ring connected to an amino group.

### 1.2.1. Dopamine

Dopamine is one of the neurotransmitters of great clinical importance for motor functions and motivational behaviors. Its dysfunction is involved in many psychiatric disorders, including drug addiction, schizophrenia, Parkinson's and Huntington's disease. The dopaminergic neurons are localized in the subsantia nigra pars compacta and in the ventral tegmental aera [52].

Dopamine seems to also be involved in drug addiction. For instance, it is known that cocaine inhibits the dopamine transporter [53]. However, cocaine can also block the norepinephrine and serotonin transporters, which can also reuptake dopamine.

Schizophrenic behavior includes a disturbance of cognition and perception with symptoms including delusions, hallucination, antisocial behavior and lack of motivation [54]. The cognitive symptoms may be caused by a hypodopaminergic activity in the mesocortical pathway, while the psychotic symptoms may be caused by a hyperdopaminergic activity in the mesolimbic pathway [55]. The role of dopamine in schizophrenia is strongly supported by the fact that dopamine antagonists are efficient to treat patients, and positive symptoms are related to dopamine dysregulation [54].

### 1.2.2. Norepinephrine and Epinephrine

Norepinephrine and epinephrine, also known as noradrenaline and adrenaline respectively, are monoamine molecules which act as hormones and neurotransmitters in the human body. As neurotransmitters, they are involved in the autonomic nervous system [56], which is composed of

the sympathetic and parasympathetic systems. The autonomic nervous system is also known as the "fight or flight" system.

In the brain, norepinehrine neurons are localized in the locus coerulus, a nucleus in the brainstem that projects to various regions of the brain, including the limbic system which is involved in the regulation of emotions and cognition [57,58]. In the hypothalamus, amygdala and locus coerulus, the increase of norepinephrine release had been associated with anxiety [59]. Adrenergic neurons are also present in different regions of the brain, including the lateral tegmental system and medulla, but the epinephrine function as a neurotransmitter is poorly understood [57]. It is believed that it plays a role in the fight-or-flight response by increasing the heart rate, vasodilatation, pupil dilation, and blood sugar [60,61].

One of the major hypotheses for depression involves a deficit in monoamines, more precisely in serotonin and/or norepinephrine [58], and the antidepressant drugs currently used act on mechanisms involving monoamines regulations. Those mechanisms are: inhibition of reuptake of serotonin/norepinephrine, antagonism of presynaptic inhibitory serotonin/norepinephrine receptors and inhibition of monoamine oxidase [58]. More evidence exists on the role of serotonin at the origin of depression, but the evidence also shows a role for epinephrine. For instance, the locus coerulus has projection to the limbic system, including the amygdala, hippocampus and hypothalamus, which are all involved in emotion and cognition. Studies on animals have also shown that increasing norepinephrine level induces a protection from depression, while norepinephrine depletion studies have shown increased risk for depression. Drugs that act specifically on norepinephrine are effective to treat depression [58].

### 1.2.3. Serotonin

5-hydroxytryptamine (5-HT), most commonly known as serotonin, is a neurotransmitter that plays an important role in many behavioral functions [62]. It regulates sleep and wake states, feeding behavior, aggressive behavior and mood/depression among others [15,57,62]. Many currently used antidepressants act on the mechanism of serotonin or norepinephrine transmission. The use of selective serotonin reuptake inhibitors has given more insight on the role of serotonin in depression [63]. Serotonergic neurons are localized in the raphe nuclei and their axons project into the prefrontal cortex, basal ganglia, hippocampus, hypothalamus and spinal cord [15,64]. A serotonin dysfunction, along with the dopamine model of schizophrenia, was proposed by Meltzer [65].

### 1.2.4. Histamine

Histamine is a signaling molecule involved in different physiological functions, and it acts as a neurotransmitter in the central nervous system. Histaminergic neurons are localized in the tuberomammillary nucleus of the hypothalamus and project to various regions of the brain and spinal cord, including the amygdala, cerebral cortex, substantia nigra, striatum, thalamus and retina [57,66–68]. Studies have shown that this neurotransmitter is involved in various disorders including Alzheimer's disease and schizophrenia [67]. Histamine may have an anticonvulsive role during development [68].

### *1.3. Other Chemical Substances*

Several other substances exist that are identified as neurotransmitters. Acetylcholine is the most studied as it is the first neurotransmitters to have been identified, due to its release at neuromuscular junctions and hence relative ease of identification and tracing [69]. It is responsible for muscle contraction at the neuromuscular junction and is also released by the post-ganglional neurons in the parasympathetic system [15,57]. In the central nervous system, the cholinergic system plays a major part in consciousness [70]. Cholinergic neurons are present in various parts of the brain and brain stem including the striatum, cranial nerves and vestibular nuclei.

An acetylcholine deficit in the cortical cholinergic system has been associated with hallucination in Lewy bodie's dementia [71]. Hypocholinergic activity in the hippocampus and in the cortex has also been associated with memory dysfunction in Alzheimer's disease [70]. Acetylcholine also plays a role in the sleep–wake cycle [72].

Other substances identified as neurotransmitters in the brain include purines such as adenosine triphosphate (ATP) [73]; some soluble gases known as gasotransmitters [74] including carbon monoxide [75], nitric oxide [76] and hydrogen sulfide [77]; and neuropeptide Y and substance P of the neuropeptide group [57].

### 1.4. Concluding Remarks on Neurotransmitters

Several chemical substances have been identified as neurotransmitters and others have yet to be discovered. As reported in Table 1, the concentration of these substances in the brain is very low and they are often only produced in specific regions; for example, L-aspartate is mostly produced in the hippocampus while dopamine is produced in the hypothalamus and substantia nigra. Nevertheless, they are among the most important substances that the body produces. They are involved in many vital functions such as vision, learning, sleep and emotions among others, and their dysfunction can lead to serious health issues, including anxiety, depression, schizophrenia, Parkinson's and Alzheimer's diseases among others. Given their important roles in the central nervous system and the severity of disorders linked to their dysfunctions, the accurate detection and measurement of neurotransmitters are among the most important challenges in neuroscience.

Given the importance of their detection, several articles reviewed the advances and challenges in neurotransmitter detection. Wu et al. [1] reviewed analytical and quantitative in vivo neurotransmitter monitoring techniques based on electrochemical and imaging approaches. Cross-species PET (positron-emission tomography) imaging advances and trends for neurotransmitter detection in brain have been surveyed by Finnema et al. [5]. Electrochemical sensing based on nanomaterial has been extensively surveyed by Sanghavi et al. [7]. The microelectronics interface design consideration was summarized in a review by Mirzaei and Sawan [9]. Advances in nanomaterials and polymer-based sensors and their potential for in situ optical neurotransmitters detection application were reported in a review by Soleymani [10]. Other works detailed the detection methodology for a specific neurotransmitters detection technique have been reported [8,11,12,14]. However, all those reviews lack a comparison of the strengths and weaknesses of the surveyed techniques. This work aims to provide such as comparison and highlight the guidelines for choosing a given neurotransmitter detection technique. The remaining of this article reviews the different methods used for their sensing and detection. It includes classical methods such as microdialysis and analytical chemistry-based methods, as well as more recent research techniques such as optical and in vivo electrochemical sensing for a faster response, instrument miniaturization, high sensitivity and selectivity. The classical techniques such as microdialysis are well studied and can be used for neurotransmitter sampling for clinical purposes in hospitals. On the other hand, the cytotoxicity of the materials used by the other techniques such as nano-object optical sensing methods is not fully studied. Thus, despite their high sensitivity and selectivity, these techniques are currently only used for in vitro research.

**Table 1.** Summary of major neurotransmitters.

| Name | Formula | Localization/Concentrations * | Role and Pathology |
|---|---|---|---|
| Glutamate | | Widespread in brain and spinal cord./ 1 µM [78], 1.4 µM [79] | Involved in learning, memory, vision, epilepsy, schizophrenia, excitotoxicity |
| L-aspartate | | Hippocampus. / 0.3 µM [78]. | Activate NMDA receptor. Co-neurotransmitter with glutamate |
| GABA | | Hypothalamus, cerebellum, spinal cordo, olfactory bulb and retina./ 0.2 µM [78], 0.17 µM [79] | Effects augmented by alcohol and antianxiety drugs, epilepsy, convulsions |
| Glycine | | Brain stem, spinal cord and retina./ 1.6 µM [78], 6 µM [79] | Hyperexcitability, uncontrolled convulsions |
| D-serine | | Striatum/ 28 µM [79], 15 µM [79] | Coagonist of NMDA receptor, schizpohrenia. |
| Dopamine | | Hypothalamus, substantia nigra of midbrain./ 26 nM [80], 40 nM [81] | Good feeling, Parkinson's disease and schizophrenia |
| Epinephrine | | Tegmental and medulla | Fight-or-flight response |
| Norepinephrine | | Locus coeruleus of the midbrain, brain stem, limbic system, cerebral cortex, thalamus./ 12 nM [82] | Good feeling, depression |
| Serotonin | | midbrain, hypothalamus, limbic system, cerebellum, pineal gland, spinal cord./ 70 nM [81], 68 nM [82] | sleep, appetite, nausea, headaches, regulation of mood, schizophrenia, anxiety and depression. |
| Histamine | | Hypothalamus | Act on on G-protein coupled receptors. Involved in Alzheimer's and schizophrenia. |
| Acetylcholine | | Basal nuclei and cortex, neuromuscular junctions./ 0.4–4 nM [83], 2.8 M [84] | Prolonged effects lead to tetanic muscle spasms, it is linked to Alzheimer's |
| Substance P | | widespread in brain, hypothalamus, limbic system, pituitary gland and spinal cord./ 23.5 pM [85] | Natural opiate |
| Neuropeptide Y | ** | Hypothalamus | Increasing food intake, reduce anxiety and pain, affect the circadian rhythm and control epileptic seizures. [86,87] |
| Adenosine triphosphate | $P_3O_{10}^{4-}$ | Basal nuclei, dorsal root ganglion./ 0.5–10 µM [88] | Involved in pain sensation |
| Carbone monoxide | $C \equiv O$ | brain, neuromuscular and neuromuscular synapses | regulates vasopressin neuronal activity |
| Nitric oxide | $N \equiv O$ | brain, spinal cord and adrenal gland./ 92 nM [89] | Relaxing factor, involved in myocardial infarction |
| Hydrogen sulfide | | Hippocampus, Hypothalamus [90] | Involved in the regulation of vascular tone, myocardial contractility, and insulin secretion |

** Neuropeptide Y IUPAC condensed notation: H-Tyr-Pro-Ser-Lys-Pro-Asp-Asn-Pro-Gly-Glu-Asp-Ala-Pro-Ala-Glu-Asp-Leu-Ala-Arg-Tyr-Tyr-Ser-Ala-Leu-Arg-His-Tyr-Ile-Asn-Leu-Ile-Thr-Arg-Gln-Arg-Tyr-OH. * *Concentrations as measured in rat.*

## 2. Neurotransmitter Detection Methods

### 2.1. Nuclear Medicine Tomographic Imaging

#### 2.1.1. Positron Emission Tomography (PET)

PET is a non-invasive neuroimaging technique that indirectly measures neuronal activity or neurotransmitters release [91,92]. In nuclear medicine, it is used to measure the metabolic or molecular activity of an organ in three dimensions, after the intravenous injection of a radioactive tracer, usually based on an injected isotope with short half-life, whose disintegration results in positron emission. Then, when a positron encounters an electron, their annihilation produces two photons of 511 keV (gamma photons) that propagate in opposite directions towards a scintillator for detection, hence defining a line of response. Each annihilation that takes place is localized by measuring the respective times of flight of the two photons before detection, and the results contribute to the reconstruction of an image of the overall process by tomography.

For cardiac, brain and tumor imaging, the radioactive element is often incorporated in glucose molecules to form a tracer with similar biological activity. Once in the blood stream, the tracer binds to glucose-consuming tissues, and the ensuing metabolic activity is recorded via functional images that are acquired by a gamma camera and reconstructed by a computer algorithm. For neurotransmitter release, the measurement relies on the competition between radio labeled neurotransmitter receptor ligands and the already bounded neurotransmitter target to its receptors [93]. It is related to measuring the change of binding potential of the receptor. For example, drug injection induces a dopamine change that is often measured with the $^{11}$C-raclopride tracer, radio labelled as a D2/D3 receptor antagonist [93–96]. Then, the dopamine release is inferred by the decrease in radioligand binding to the receptor sites.

PET offers the potential of accurately finding where a given neurotransmitter is concentrated in the brain, but since it is based on the tracking of the tracer location, and the emitted photons are attenuated differently throughout the body, it is often necessary to normalize the observed activity with sophisticated algorithms for improved detection. On the other hand, the technique is almost with no risk to the patient, since a very small dose of radioactive tracer with low half-life is used.

Before 2009, the so-called Novel Methods leading to New Medications in Depression and Schizophrenia (NEWMEDS) only validated radioligands for endogenous dopamine detection. Since then, the development of new radioligands permit the detection of extracellular concentrations of many neurotransmitters including dopamine, serotonin and noradrenaline in human, monkey and rat brain [5]. Indeed, the combination of PET, MRI and EEG are now used in clinical domain for increased spatial and temporal resolution measurements of neurotransmitters such as serotonin [6].

#### 2.1.2. Single-Photon Emission Computed Tomography (SPECT)

SPECT is another nuclear tomography neuroimaging technique that can be used to observe neurotransmitter release in the human brain [97,98]. To perform a SPECT, a radioactive product that emits gamma radiation is injected into the patient bloodstream. This radiotracer is chosen according to its chemical properties to selectively bind to some tissues and thus highlight certain biological processes. Then, a gamma camera is rotated around the patient to create an image of the radiopharmaceutical distribution. The camera consists essentially of a collimator that allows the angular selection of photons and gamma-ray detectors, and a tomographic reconstruction algorithm estimates the three-dimensional mapping of the radioactive activity, hence giving the distribution of the radiotracer in the body.

Like PET, SPECT is based on the in vivo competition between the endogenous neurotransmitter and the radiotracer to bind to a receptor. For example, the stimulation of dopamine release by amphetamine reduces the binding of $^{123}$I-iodobenzamide ($^{123}$I-IBZM), which is a dopamine antagonist. However, SPECT differs from PET in other aspects: (1) the single gamma photon emitted in SPECT comes directly from the radioisotope atom, as opposed to the two photons created by annihilation

in PET [99], hence allowing PET to offer higher spatial resolution (as it uses two photons to localize each point of interest); (2) More massive isotopes, such as [123]I-IBZM, can be used to label other molecules. For instance, [123]I-IBZM is a used in SPECT to study dopamine neurotransmission in the human brain [97,98]; (3) SPECT exhibits a much lower sensitivity (approximately two to three orders of magnitude) than PET, but it is also less costly.

While SPECT and PET are successful techniques for the spatial visualization of molecules such as neurotransmitters, they are limited by the complexity involved in label development, due to the signal overlapping between the labeled compound and the target metabolite [2]. The matrix-assisted laser desorption/ionization mass spectrometry (MALDI-MS) imaging was developed to provide a high resolution multiplexed spectra at near-cell spatial resolution [3]. However, applying MALDI-MS requires the modification of most neurotransmitters due poor ionization efficiency. Shariatgorji et al. used pyrylium salts to ionize several neurotransmitters and, in combination multiplexed data acquisition, several neurotransmitters including GABA, glutamate, serotonin and dopamine, among others, could be accurately detected [2].

### 2.2. Optical Sensing Techniques

#### 2.2.1. Surface-Enhanced Raman Spectroscopy (SERS)

SERS uses photon interactions with matter to detect and identify molecules. This Raman spectroscopy technique enhances the Raman scattering by molecules adsorbed on an irregular metal surface, with an amplification factor that can reach 10 to 100 billion [100]. The method requires a functionalized surface for each molecule, and it can be made sensitive enough to detect single molecules [101]. The signal is enhanced by two main effects: electromagnetic and chemical enhancements. The enhancing materials are usually made of noble metals, e.g., Au/Ag nanorods and Au/Ag nanoparticles, and they are used as substrates to generate high electromagnetic fields. However, the SERS signals derived from individual nanoparticles are often too weak for detection; for high SERS signals, nanoparticle assemblies with tunable gaps can improve the electromagnetic fields. In general, the smaller the gap, the stronger the signal [102].

Suitable SERS substrates were designed to detect various neurotransmitters by adding silver colloids [103], silver nanoparticles [104] or hollow core photonic crystal fiber [105] to the cell culture medium. SERS is used to detect concentrations of choline (Limit Of Detection (LOD): 2 μM), acetylcholine (LOD: 4 μM), dopamine (LOD: 100 nM) and epinephrine (LOD: 700 μM) [104], glutamate (LOD: 100 nM) and GABA (LOD: 100 μM) [105]. In the case of dopamine, an LOD of 6 fM was achieved using Au@Ag nanorod dimers based on aptamer that are thiolated, which DNA specifically binds to dopamine [102].

Although noble metallic nanorods and nanoparticles coupled with aptamers permit a highly sensitive and selective SERS, some of them are toxic or cause irreversible aggregations. Thus, techniques such as the one described in [102,104,105] are usually not usable for in vivo diagnosis.

#### 2.2.2. Fluorescence

Fluorescence is the emission of photons by a molecule or material such as metal nanoparticles or quantum dots, following light stimulation at one or more specific wavelengths. It reflects the returns of electrons to their ground state after shifting into a higher energy state by the incident light. The electrons dissipate part of the received energy through vibrations and return to their ground state by emitting a photon with the remaining energy, which is typically lower than the incident one, yielding a longer emitted wavelength. For example, neurotransmitters including serotonin, dopamine and norepinephrine have been detected by fluorescence measurements carried out at 320 nm, with excitation at 279 nm [106].

Although the fluorescence sensing method is sensitive, interfering compounds can seriously hinder its neurotransmitter detection efficiency. For example, 5-methoxytryptamine, N-acetylserotonin

or melatonin all affect serotonin detection [107]. Solvent extraction with few interfering materials may be avoided with ion exchange resin (Amberlite CG-50), which acts as a suitable vehicle for the refinement and concentration of serotonin from biological samples [108]. In addition, high-sensitivity fluorescence sensing of serotonin was achieved with ninhydrin, which acts as a fluorescence catalyst [109]. Fluorescence efficiency was also improved through the use of O-pthaldehyde (OPT) under heat and strong acid conditions [110]. This method requires only 50 mg of tissue samples, but the extraction, concentration and purification methods, and the assay method as well, are time consuming and the fluorescence intensity is low. Serotonin can also be detected in biological fluids using ethanol extraction, which enhances fluorescence intensity without any time-consuming steps [111]. However, this technique is not appropriate for repetitive analysis as serotonin in blood is rapidly oxidized by oxyhaemoglobin [111]. Adding ascorbic acid with dry ice acetone can avoid this oxidation [112]. Finally, combining fluorometric detection with HPLC reached a high sensitivity and selectivity for serotonin [113] and dopamine [114] with an LOD of 1 pM concentration, and a signal to noise ratio of 5.

An in situ fluorescein-based neurotransmitter detection protocol was developed by Dey et al. for the rapid and low-cost rapid detection of histamine. Dye-coated paper strips were used to detect histamine in various biological fluids, including human plasma and urine. The spectral analysis of the samples revealed that this technique can be used to detect histamine at concentrations as low as 24 nM [4].

Fluorescent single-walled carbon nanotubes have been used to detect dopamine with an LOD of 11 nM [115]. The single-walled carbon nanotubes are modified with a DNA/RNA coronae that acts as a conformational switch, thus reversibly modulating the nanotube fluorescence. As reported by Kruss et al., by using a specific polymer wrapper, this technique can be used to detect other catecholamines with very high sensitivity and selectivity [115]. A similar work was reported by Mann et al. with an improved LOD of 2.3 nM [116].

### 2.2.3. Förster Resonance Energy Transfer (FRET) and Photoinduced Electron Transfer (PIET)

FRET is based on the radiation-free transfer of energy from a "donor" fluorophore to an "acceptor" fluorophore [117,118]. Upon excitation, the electrons at the valence band of the donor move to a higher energy or conduction band, and then immediately return to the lower energy state. Thus, an emission of light at different wavelengths is observed [119]. The quenchers (acceptors) absorb the energy from the donors in the excited state. Consequently, an energy transfer and subsequent quenching of the donor fluorophore is observed, while the acceptor fluorophore sees an increase in its emission intensity [120]. The quenchers are normally organic dyes, quantum dots (QDs), nanoparticles or biological molecules with an absorption spectrum close to the emission spectrum of the donor.

For instance, dopamine can quench the fluorescence of QDs [121–123]. Quinone, an oxidized form of dopamine, is considered as an electron acceptor. When the QDs are excited to a higher energy state, their electrons are pushed to the lowest vacant molecular orbital of quinone and, then, the electrons which are transferred back to the valence band of the quantum dot induce a quenched state of QDs. The rate of electron transfer depends on the pH and concentration of the oxidized dopamine in the solution [124]. QDs can transfer energy to molecules able to generate singlet oxygen [125]. Dopamine can thus behave as a photosensitizer by generating singlet oxygen through auto-oxidation. In alkaline solutions, functionalized quantum dots are conjugated to dopamine via hydrogen bonding which can be oxidized to quinone, leading to quenching the fluorescence of the quantum dots with a high LOD of 0.3 nM [122,126,127]. Hybrid quantum dots with natural thiol containing molecules on the surface have good solubility and high selectivity for dopamine in biological fluids, for a high LOD of 0.8 nM [128].

As cadmium-based quantum dots are expensive and toxic, they have been replaced by more affordable ZnO quantum dots that are biocompatible and environmentally-friendly for dopamine sensing [129]. Fluorescence detection techniques without heavy metal ions release were developed

through the use of natural amino-acids as surface coating to show a high LOD of 875 pM [130]. Considering the heavy metal toxicity and environmental hazards [131], graphene quantum dots are preferred to semiconductor based quantum dots due to their environmentally-friendly properties [132], excellent water stability (neurotransmitters are always in aqueous solutions), stable photoluminescence and high electrical conductivity [133]. Polyprole microsphere–graphene core–shell hybrid systems for dopamine open a new prospect for sensing dopamine with LOD of 10 pM in real-time sensing [134].

Gold nanoparticles are also excellent acceptors to replace the traditional organic quenchers [135] for photoinduced electron transfer based dopamine sensing [136]. Gold nanoparticles can be functionalized with aptamers, which are single-stranded nucleic acids with a specific affinity to one molecule—for example, dopamine. They are screened by an in vitro selection technique named SELEX (systematic evolution of ligands by exponential enrichment) [137,138]. Dopamine binding aptamers prevent the aggregation of gold nanoparticles upon NaCl addition. Following the addition of dopamine, it is conjugated to the dopamine binding aptamer, leaving unprotected nanoparticles . The later aggregate, resulting in the fluorescence quenching of rhodamine B. The fluorescence intensity is proportional to dopamine concentration with an LOD of 2 nM [139].

### 2.2.4. Chemiluminescence

Chemiluminescence is the production of light during a chemical reaction: An energy donor molecule enters an excited state and transfers that energy to an acceptor (luminescence carrier), which releases a photon in order to return to its fundamental state, hence the luminescence. The technique was used for the identification of catecholamines in the culture of a clonal line of rat pheochromocytoma cells [140]. The chemiluminescence generated with the reaction of bis(2,4,6-trichlorophenyl)oxalate and $H_2O_2$ was used in an HPLC (High Performance Liquid Chromatography, described below) detection system to determine fluorescamine-labeled catecholamines [141]. The technique was coupled with HPLC for the separation of various biological samples with high selectivity [142]. Other works used a flow injection inhibitory chemiluminescence to detect the concentration of dopamine hydrochloride [143,144]. The method relies on the ability of dopamine hydrochloride to inhibit the chemiluminescence of a luminol–potassium hexacyanoferrate(III) system [145,146] and a potassium permanganate–formaldehyde system in acidic medium [147]. A sensitivity of 6 nM was reported [143].

The real-time measurement of catecholamines in urine samples was achieved by combining HPLC and flow injection with the chemiluminescence generated through the Ca inhibition effect on the oxidation of luminol by iodine in alkaline solution [144]. Further advancement in flow injection systems with plant tissue-derived molecular probe polyphenol oxidase is also used [148,149], catalyzing the oxidation of dopamine by oxygen. The latter generates hydrogen peroxide to induce chemiluminescence with luminol [150].

### 2.2.5. Optical Fiber Biosensing

The principle of optical fiber biosensors is based on the optical transduction induced by the binding of the target molecule to an immobilized receptor on optical fibers acting as transduction elements [151]. The optical fiber core is doped with germanium for a higher refractive index than the surrounding cladding made of silica, which then induces light propagation through total internal reflection [152]. When light passes through the optical fiber, the generated electromagnetic field or evanescent wave is related to the distance-dependent, rapid light intensity decay that occurs between the interface and the medium of smaller refractive index. In the case of molecular recognition, the evanescent light excites a fluorophor and the resulting fluorescence emission is propagated through the fiber towards an optical sensor [153]. The optical fiber-based sensing technique was first applied to glutamate sensing by immobilizing glutamate dehydrogenase at the probe (core), which catalyzes the production of reduced nicotinamide-adenine dinucleotide (NADH) upon oxidation with glutamate. The obtained NADH produces fluorescent light that is proportional to the

concentration of glutamate [154]. This technique is advantageous in monitoring glutamate release and continuous monitoring of neurophysiological responses, and an LOD of 0.22 μM and absolute mass detection limit of 44 fg were reported [155].

Fiber optic aptasensors have been used for dopamine detection around the physiological concentration (LOD: 37 nM) [156]. The sensor is composed of a recognition element made of a 57-mer dopamine binding aptamer and a probe made of a nonadiabatic tapered optical fiber. The aptamer has a specific affinity to dopamine and, in the presence of dopamine, it changes its structure from a random conformation to a rigid pocket-like tertiary conformation as shown in Figure 2. This change of tertiary conformation induces a variation of the refractive index around the tapered fiber, hence indirectly permitting the detection of dopamine.

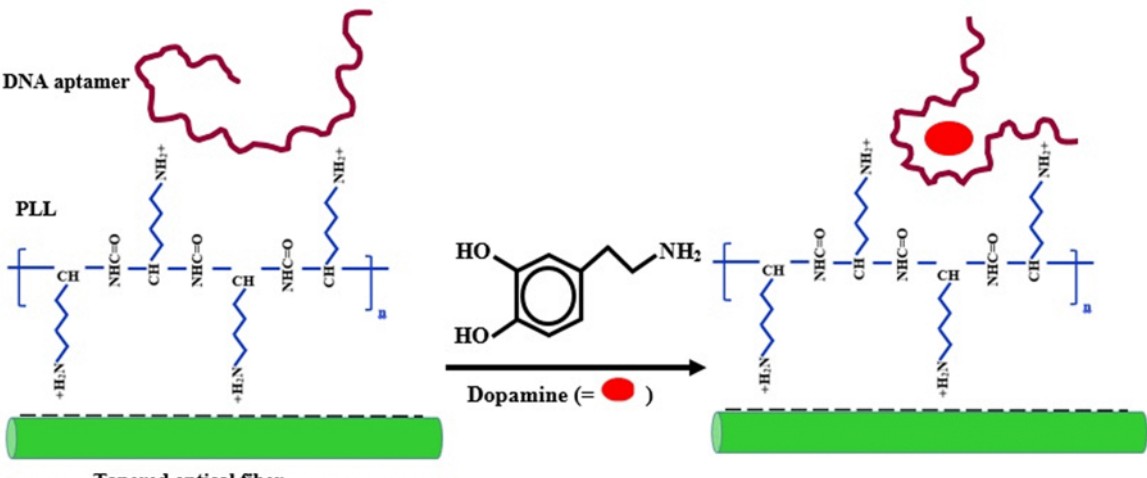

**Figure 2.** Schematic representation of the fiber optic and aptamer based dopamine detection [156].

2.2.6. Colorimetry

This technique uses the interaction between a molecule of interest and other compounds to induce a colored reaction. The concentration of the target molecule then depends on the color and intensity of the resulting solution. Using this technique, dopamine was detected when binding with Ag-catechol [157], gold nanoparticles [158] and AHMT functionalized gold nanoparticles [159]. The reported LOD was 60 nM [157]. Other neurotransmitters were also detected using this technique, such as noradrenaline (LOD : 20 μM), adrenaline (LOD: 2.5 μM) and L-DOPA (LOD: 2.5 μM) [160].

The colorimetry technique can also use resonance light scattering. It then induces a collective oscillation of electrons on the surfaces of plasmonic nanoparticles and results in an absorption/scattering effect at different wavelengths upon aggregation. The resonance light scattering technique can be advantageous due to its sensitivity and selectivity. Plasmonic nanoparticles based on gold and silver are highly efficient at light absorption and scattering [161]. For example, dopamine has been detected though a shift in wavelength induced by the aggregation of gold nanoparticles synthesized primarily with dopamine as reducing agent [160,162,163]. The indirect aggregation of nanoparticles is caused by the hydrogen bond formation within the Au-dopamine-TGA complex, thereby inducing a shift from red—purple—gray—yellow when dopamine concentration increases. However, a reversible shift (blue to red) was induced by dopamine concentration in aggregated melamine functionalized gold nanoparticles [164,165]. Furthermore, aggregation induced by the selective binding between gold optical probes and dopamine through specific molecular bridges including aptamers avoids common interference with other neurotransmitters [159,166]. In some occurrences, dopamine sensitivity is affected by the medium because, it is oxidized to quinone which is unstable and quickly polymerized to polydopamine in alkaline conditions. However, polydopamine nanoparticles are intrinsically fluorescent in an acid medium, allowing dopamine sensing with an

LOD of 40 nm [167]. The reliable detection of dopamine with the low-cost resorcinol, reaching an LOD of 1.8 nM, is a good alternative for the real-time detection of this neurotransmitter [168]. Although neurotransmitter detection in the cerebral system is still challenging, the colorimetric technique to detect dopamine with functionalized gold nanoparticles is promising [158].

## 2.3. Electrochemistry

Electrochemistry-based detection relies on measuring electrical quantities such as current, potential or charge in relation to chemical parameters [169]. It has been extensively used for the study of catecholamines [42], due to their oxidation properties. In vivo measurement of catecholamines in extracellular fluid was achieved with carbon microelectrodes implanted into the brain of a rat [170].

The direct detection of neurotransmitters in the brain by this technique is very challenging given their very small concentrations, notwithstanding the presence of interfering compounds [171]. For example, the oxidation of ascorbic acid occurs at a voltage that is close to those of catecholamines, thus making their detection in the presence of ascorbate difficult [172,173]. Next, a description of the three main electrochemistry detection techniques is presented: fast-scan cyclic voltammetry, amperometry and coulometry.

### 2.3.1. Fast-Scan Cyclic Voltammetry (FSCV)

Voltammetry relies on measuring the current flowing between two electrodes, a working electrode and a counter electrode, subjected to a voltage difference during a redox chemical reaction. The working electrode is responsible for the redox current and the counter electrode maintains a constant potential while passing a current to counter the redox events at the working electrode. In practice, it is difficult to make this set up work and a third electrode, called the reference electrode, is added to relieve the counter electrode of maintaining the voltage difference.

In FSCV, the applied voltage is a repetitive ramp with a high slew rate [174]. During FSCV scans, an electroactive neurotransmitter near the working electrode rapidly oxidizes and transfers electrons to the electrode surface, thus generating a faradic current [175]. A reduction faradic current of approximately the same order as the oxidation current is also observed [174]. FSCV can be based on carbon-fiber electrodes to detect indolamines such as serotonin, and catecholamines such as adrenalin, noradrenalin and dopamine as illustrated in Figure 3 [176], since they can be oxidized at low voltage [177]. FSCV can be used for in vitro as well as in vivo experiments in single neurons, brain slices, and anesthetized or moving animals such as a rat with a small headstage fixed on the head [178].

When used for neurotransmitter sensing, FSCV consists first of maintaining the electrode potential beneath the oxidizing potential of the neurotransmitter of interest (e.g., –0.4 V for dopamine with an Ag/AgCl reference electrode). Then, the electrode potential is increased above the neurotransmitter oxidizing potential (around 1.0 V) at a high scan rate, usually 400 V/s. The duration of each cycle is 100 ms [177,179,180], and the measured current depends on the neurotransmitter concentration.

The main difference between FSCV and Cyclic Voltammetry (CV) is the very high scan rate used in FSCV. Indeed, the scan rate can be much higher than 400 V/s and the duration of each cycle can vary depending on application. It has been reported that FSCV is more convenient for neurotransmitter sensing in in vivo experiments because it has a better resolution than CV. However, both techniques should be able to detect the same molecules. If the molecule is not detectable with CV at high concentration, we should not expect it to be detected through FSCV either.

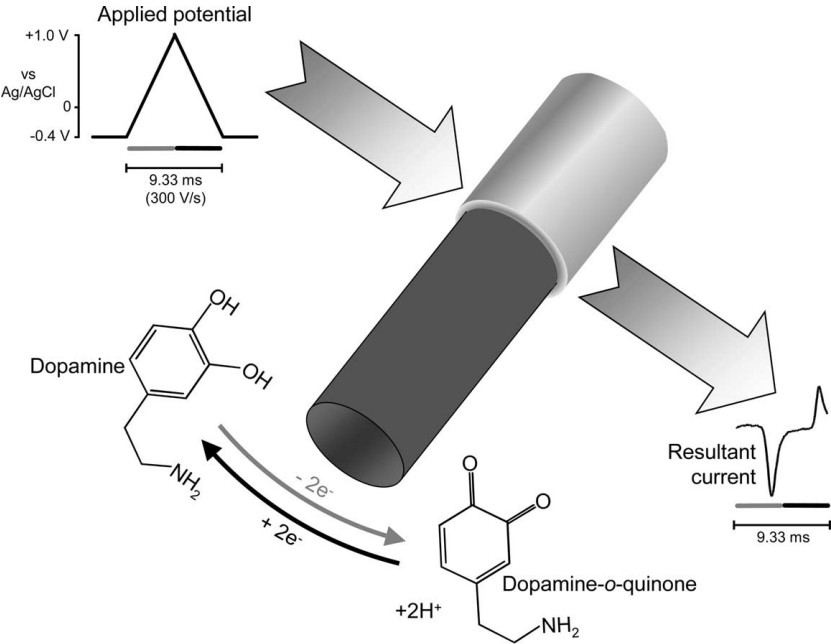

**Figure 3.** Voltammetric detection of dopamine. Dopamine is oxidized to dopamine-o-quinone when a potential is applied to the electrode. The resulting electrons from this oxidation generate a current that is detected. When the potential is removed from the electrode, the dopamine-o-quinone is reduced back to dopamine, thus producing a current in the reverse direction [179].

Each voltammogram (current vs. voltage curves) is unique to a substance and can be used to identify neurotransmitters as illustrated in Figure 4 [180]. Indeed, the voltammogram depends on the molecule's properties such as its electron transfer rate and chemical stability [181]. Those factors determine the positions and shapes of the reduction and oxidation peaks, as well as their relative amplitudes [181]. As the current generated at the oxidation voltage corresponds to the concentration fluctuation versus time, it represents the changes in neurotransmitter concentration upon external stimulation [182]. FSCV based on in vivo dopamine sensing in the rat brain was demonstrated with Nafion-coated microelectrodes [183], tungsten-coated microelectrodes [184] and carbon fibre microelectrodes [183] to measure sub-micromolar changes in the concentration of dopamine upon stimulus. To enhance the sensitivity and selectivity of this technique for dopamine, electrode surface functionalization is needed such as modified carbon paste solid electrodes [185,186] and electrically-treated carbon micro-electrodes [187]. Electrode functionalization with polymers was also investigated, such as Nafion-coated electrode which is specific to cationic species such as dopamine, serotonin [188], clay-modified electrodes [189], electroactive polymer such as polypyrrole coated [190], poly (N,N-dimethylaniline)-modified electrodes [191], and uncoated carbon fiber microelectrodes [192]. A higher sensitivity is achieved by extending the anodic scan rate to +1400 V/s. Then, a brain tissue dopamine detection limit of 50 nM was achieved [193]. However, the detection of dopamine and serotonin is problematic because of the reactive species that are generated after oxidation and stick to the electrode surface, thus affecting the system sensitivity [194,195]. Therefore, when dopamine and serotonin are mixed together, selectivity becomes a challenge due to the similarities of the two molecules. That limitation can be improved by using carbon nanotube-modified microelectrodes [196]. Furthermore, selective dopamine detection in the presence of ascorbic acid was reported through electropolymerization of polyglycine on carbon paste electrodes, which improves the redox peak currents of dopamine [197].

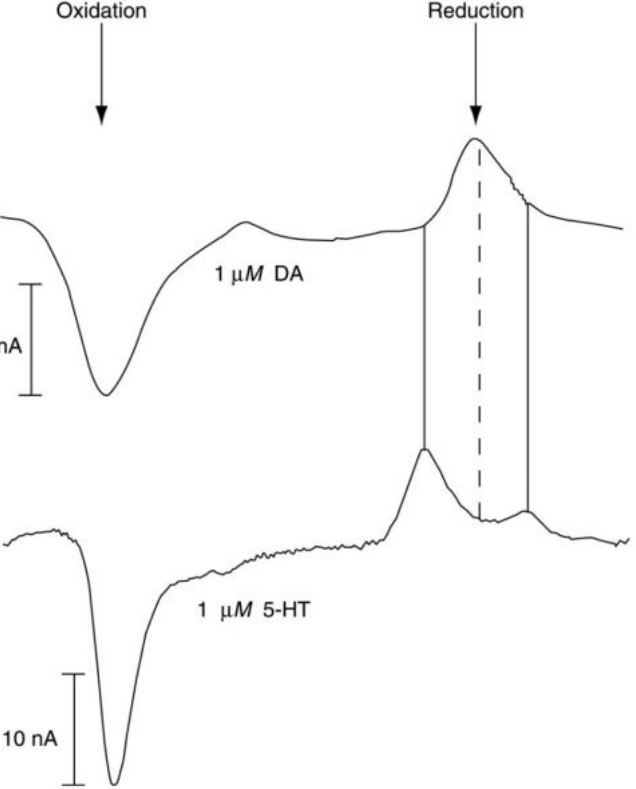

**Figure 4.** Dopamine (DA) and serotonin (5-HT) voltammograms. The voltammogram of dopamine 1 μM shows one oxidation and one reduction peak. The voltammogram of serotonin 1 μM shows one oxidation and two reduction peaks [177].

### 2.3.2. Amperometry

Amperometry applies a fixed potential to a working electrode and measures the resulting oxido-reduction reaction current, which is proportional to the analyte concentration [198]. Like voltammetry, the current through the working electrode is measured with respect to a reference electrode, but, unlike FSCV, the temporal resolution of amperometry is not limited to the duration of the cycle. Thus, amperometry is more suitable to analyze the kinetics of neurotransmitter release.

The catecholamines and indolamines in a cell are efficiently detected with amperometry. A carbon-fiber microelectrode maintains a constant voltage above the oxidizing potential of the neurotransmitter (Figure 5), which is 650 mV and 800 mV for serotonin and catecholamine, respectively. The electrons released by the neurotransmitters are then transferred to the electrode [179,199]. The temporal resolution is limited by two factors: the diffusion of the neurotransmitter to the electrode and the kinetics of the electron transfer [180]. The current measured by the electrode is dependent on the concentration of neurotransmitter released by the cell [200], but the released neurotransmitter cannot be identified [179]. In this case, a complementary method such as high performance liquid chromatography is needed [200].

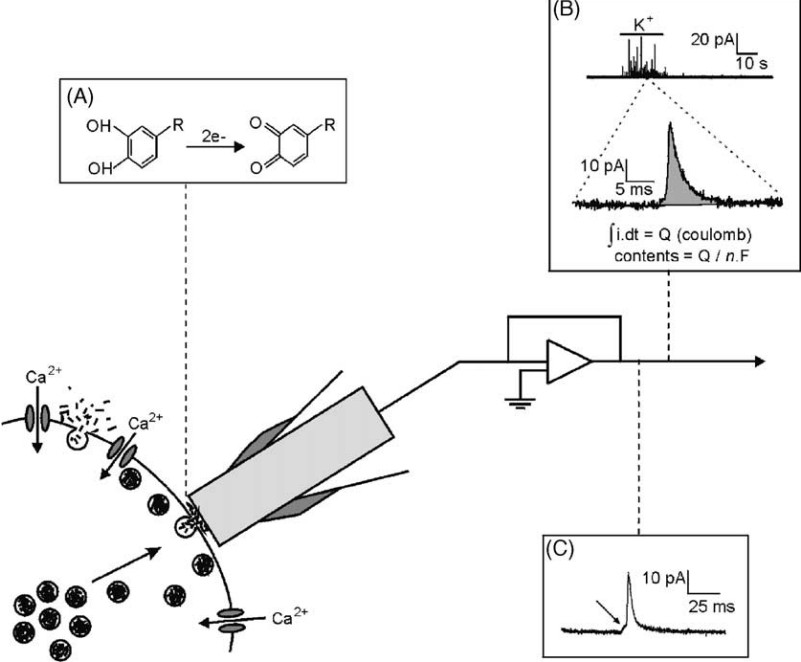

**Figure 5.** Detection of dopamine by amperometry. Experimental set-up for amperometric recording of vesicular catecholamine release from a secretory cell using a carbon fiber microelectrode; (**A**) schematic drawing of the electrochemical reaction; (**B**) amperometric recording from a PC12 cell; (**C**) example of an amperometric current transient on an expanded time scale [200].

## 2.4. High Performance Liquid Chromatography (HPLC)

HPLC separates the different molecules in a mixture called the mobile phase, which is injected into a separation column of a diameter of typically 3.0–4.6 mm, initially filled with a stationary phase made of porous particles of 2.5–10 μM diameter [201,202]. The molecules are introduced in the mobile phase under high pressure and then move through the stationary phase. The Van Deemter equation describes the relationship between linear velocity (flow rate) and column height [201]. Depending on the molecules' properties, the interaction with the stationary phase is different: the molecules having a high affinity with the mobile phase migrate along the column faster, and are therefore the first to pass through it. The peak on the chromatograph is observed as shown in Figure 6, and the surface beneath the peak is proportional to the molecule's quantity [201,203]. HPLC is a very sensitive technique that allows the detection of two enantiomers. it was used to detect D-serine in the brain [46]. It is also often coupled with other techniques such as chemiluminescence. When coupled with mass spectrometry, it has been used to detect several neurotransmitters including GABA, pyroglutamic acid, aspartic acid, asparagine, glutamic acid, glutamine, N-acetyl-L-aspartic acid, acetylcholine and its metabolite choline. The reported LOD for those neurotransmitters were 1.820, 0.046, 0.088, 1.930, 1.520, 1.050, 0.250, 0.012, and 0.620 nM, respectively.

Simultaneous detection of monoamine and amino acid neurotransmitters in rat endbrains was achieved by pre-column derivatization with high-performance liquid chromatographic fluorescence detection and mass spectrometric identification. Five molecules including L-glutamate, GABA, dopamine serotonine and 5-hydroxyindole acetic acid, which is a metabolite of serotonine, were separated and then detected with an LOD of 0.398–1.258 nM [204].

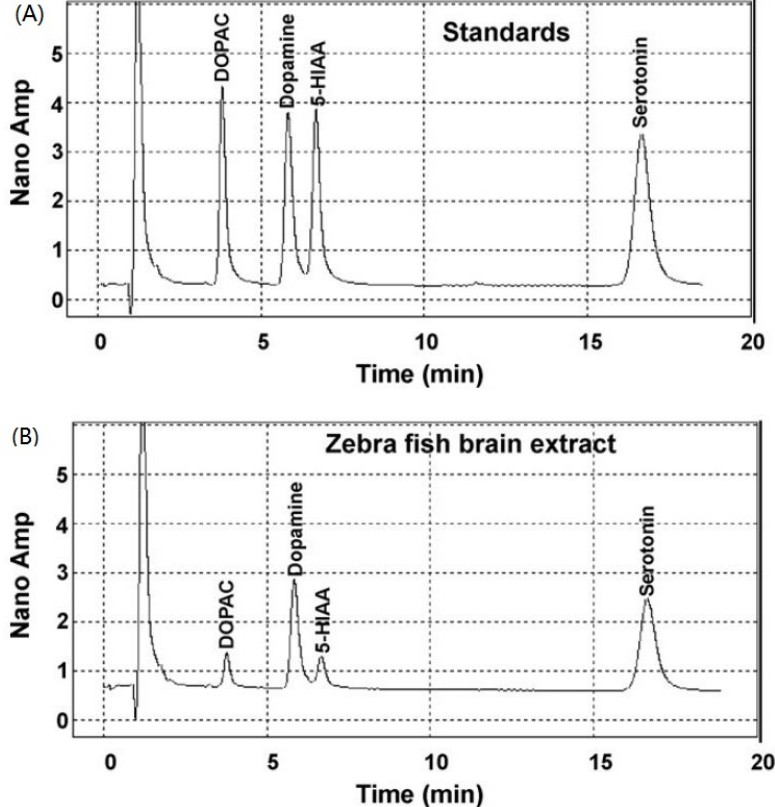

**Figure 6.** HPLC analysis of some neurotransmitters; chromatogram of neurotransmitters separated by the microbore HPLC column. (**A**): standard DOPAC (50 pg), dopamine (150 pg), serotonin (100 pg) and 5-HIAA (50 pg); (**B**): DOPAC, dopamine, serotonin and 5-HIAA separated from zebrafish brain lysate [205].

## 2.5. Microdialysis

Microdialysis is one of the well studied neurotransmitter detection techniques and its in vivo applications have been used for over three decades. For instance, it has been used for the measurement of neurotransmitters such as acetylcholine, neuropeptide, amino acids and amines in human brain [206]. It is usually used in conjunction with other detection and separation techniques such as liquid chromatography and mass spectrometry [207]. The technique consists of inserting into the brain a probe formed by a semi-permeable membrane of 1–4 mm length and 0.2–0.4 mm diameter, perfused at a constant flow between 0.1–5 μL/min as shown in Figure 7 [206,208,209]. Artificial cerebrospinal fluid is used to perfuse the probe to prevent ionic diffusion [208,209]. The membrane is semi-permeable so that molecules in the extracellular fluid will diffuse according to their gradient [206,208]. This principle can also be used to inject molecules into the extracellular fluid for reverse dialysis or retro-dialysis [208,209]. The micro-dialysis probe is placed outside the synaptic gap. Therefore, the neurotransmitter concentration measured by micro-dialysis is a balance between the molecules released into and removed from the extracellular space [208].

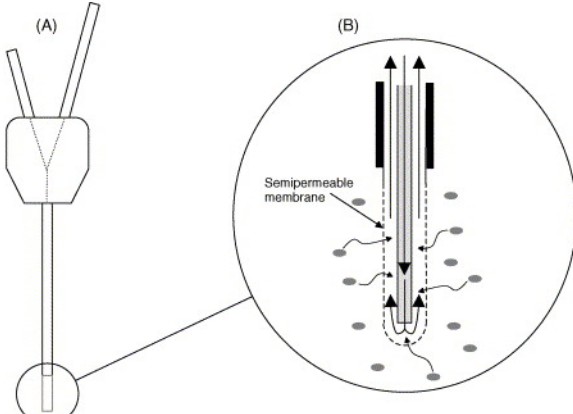

**Figure 7.** Microdialysis probe. A microdialysis probe is formed by a semipermeable membrane and perfused at a constant flow. Molecules will be able to diffuse across the membrane according to their gradient [209].

*2.6. Other Detection Techniques*

Tandem Mass Spectrometry

Mass spectrometry ionizes analytes and sorts neurotransmitters according to their mass-to-charge ratio. The Tandem Mass Spectrometry is based on the same principle but uses multiple stages of detection and fragmentation. It is often combined with various types of chromatography [210–212] to detect over twenty-five types of neurotransmitters [213].

A detection method known as ultra-high performance liquid chromatography-tandem mass spectrometry, based on derivatization techniques developed by Zhao et al., can not only determine a variety of neurotransmitters but also can bring super high sensitivity, accuracy, selectivity, and the combination of high sensitivity and high time resolution of in vivo microdialysis [214]. This technique may play an important role in fast neuropathies diagnostics. In situ ultrasound-assisted derivatization dispersive liquid–liquid microex-traction coupled with ultra high-performance liquid chromatography tandem massspectrometry was demonstrated to be a potential method for sensitive, accurate and simultaneous monitoring of amino acids and monoamines neurotransmittes for neurodegenrative disordders such as Alzheimer's disease [215].

## 3. Neurotransmitter Detection Techniques' Guidelines

The surveyed neurotransmitter detection techniques all have strengths and weaknesses as shown in Table 2. The suitability of a technique over the others depends on the intended application. For instance, in vivo usable techniques are better than those only usable in vitro for clinical applications, but this often comes with more complexity, lower sensitivity and/or selectivity, and higher cost. In addition, the methods with higher sensitivity and selectivity are often bulkier and require complex and time-consuming manipulations.

Nuclear medicine tomographic imaging offers high spatial resolution, but the equipment is expensive, making these techniques only suitable for diagnostic purposes where the functional studies and accurate localization of neurotransmitters in the brain are needed. In addition, since they are not suitable for neurotransmitter concentration detection, they must be used with techniques that have higher concentration sensibility to get accurate neurotransmitter concentration measurements. In hospitals and research laboratories, they can be used with microdialysis or analytical chemistry techniques such as HPLC.

**Table 2.** Summary of neurotransmitters' detection techniques.

| Techniques | Advantages | Shortcomings | Reported LOD |
|---|---|---|---|
| PET | High spacial resolution | Complex manipulation Very high cost | Dopamine: 200 nM [216] |
| SPECT | High spacial resolution | Complex manipulation Very high cost | |
| SERS | Very high sensitivity and selectivity | Can be inaplicable in vivo depending on used material. | Choline: 2 $\mu$M Acetylcholine: 4 $\mu$M Dopamine: 100 nM Epinephrine: 100 $\mu$M |
| FSCV | High sensitivity | Low selectivity Electrode short lifetime | Dopamine: 50 nM |
| Amperometry | Low implementation cost | Low sensitivity and selectivity | Dopamine: 10 nM [217] |
| HPLC | High sensitivity and selectivity | High cost and complex manipulation | |
| Fluorescence Chemiluminescence | High sensitivity and selectivity | May not be usable in vivo | Dopamine: 10 pM |
| Optical Fiber Sensing | High selectivity | Low sensitivity | Glutamate: 0.22 $\mu$M |
| Colorimetric | High sensitivity and selectivity, low cost | Not usable in vivo | Dopamine: 1.8 nM Noradrenaline: 20 $\mu$M Adrenaline: 2.5 $\mu$M |

Analytical chemistry techniques may require costly equipment, but, as they are traditionally used in most bio-chemical laboratories for various chemical analyses, this equipment is available for most neuroscientists. In addition to cost, their inconvenience is that they require complex manipulations and are time consuming. On the other hand, microdialysis, sometimes used in conjunction with other methods such as HPLC, is one of the most used neurotransmitter detection methods for clinical use. It is well studied and can be used for continuously monitoring drug or metabolite concentrations in the extracellular fluid of virtually any tissue. However, as with all invasive detection techniques, it presents practical and ethical limitations for most researchers. Moreover, its temporal and spatial resolution is very limited when compared to nuclear medicine tomographic imaging.

With the rapid development of innovative technologies, and the emergence of multidisciplinary research outcomes in biomedical engineering, the current research tends to focus on more compact, portable and implantable neurotransmitter sensors. As a result, several techniques have been proposed or revisited. For instance, the electrochemical detection technique is reputed to be easy to implement in an implantable device, in addition to being cost effective. However, due to its low selectivity and the complexity of in-vivo chemical/molecular monitoring, its usage in real world applications still needs improvements in terms of reliability and selectivity. In addition, not all neurotransmitters can receive or yield electrons in a redox reaction on an electrode surface. Thus, only reactive neurotransmitters such as dopamine can be detected by using an electrochemical method without electrode functionalization. The electrochemical approach is also closely dependent on the electrode surface, which reacts with the targeted molecule. Thus, it is not possible to guarantee the stability and quality of the electrode for a long time; although it is possible to detect a good signal over a small number of detection cycles, the electrode degradation becomes significant as the number of cycles grows. This affects the number of electrons that can be trapped by the electrode, hence lowering the sensed current. Consequently, in terms of engineering, it is impossible to guarantee consistent results over time. That is the reason why, for instance, in the case of glucose sensors, the commercial products are based on one-time usage electrodes in order to guarantee result reliability.

If we compare the electrochemical research for glucose sensors and brain electrochemical sensing, the environment itself is very different. The glucose sensors are adapted to blood which is a very non-homogeneous liquid, but available in important quantities, while, in the case of the brain, the opposite is true: the cerebrospinal fluid volume is available in less volume, but it is more homogeneous. Moreover, brain volume samples are less reachable than blood because of the complexity to access the sampling zone, and when an electrode is implanted into the brain, the tissue can react with

the implanted devices. All of these aspects make the use of electrochemical methods problematic for brain research. Still, some research uses this technique, whereas other techniques are being developed.

Optical sensing techniques are suitable for miniaturization, thanks to rapidly advancing optoelectronic and MEMS (micro-electromechanical systems ) micro-fabrication technologies, and the possibility to automate chemical reactions with micro-reactors. With adequately selected functionalized quantum dots (QDs), enhanced surfaces by noble metals particles or aptamers, this emerging technology is among the most promising for neurotransmitter detection with high accuracy. One of the known optical sensing weaknesses is the limitations due to Debye length. While aptamer modified surfaces provide high selectivity in SERS based biomolecule detection, Raman signal is only enhanced within the physiological Debye length, which is approximately 1 nm. Hence, this may complicate the detection of small molecules such as neurotransmitters. However, Nakatsuka et al. proposed using aptamer modified field-effect transistors (FET), which can overcome this limitation [218]. Their strategy of combining both high sensitive FET and high selective aptamer may open another branch of neurotransmitter detection based on electronics in parallel to the known optical techniques.

Another weakness of optical sensing techniques is that some of the sensing elements may be toxic and thus unusable for in vivo neurotransmitters detection. Furthermore, these techniques rely on the optical properties of neurotransmitter molecules. The approach is acceptable when experiments are performed in a laboratory environment and in vitro experiments. However, when molecules in the brain are targeted, the tissue interaction with and growth around the sensor should be considered. It is technically impossible to guarantee that the sensor environment will not be changed when in contact with the cerebral tissue. Knowing that most of optical sensors are based on optical stimulation of molecules, the optical path between the light source and the target molecule must remain clear, which is impossible to guarantee with actual technology, even if we use biocompatible materials. Indeed, when using such materials, we only ensure their non-toxicity, either directly or through secondary reactions. However, in many applications, the tissue growth helps to stabilize the system as is the case with pacemakers. Thus, we are confronted with controversial solutions: on one side, the tissue growth is important to stabilize the system in vivo, and, from the other side, it is not suitable to avoid optical path obstruction. In addition, unlike electrical techniques where electrical signals can pass through the tissue when the amplitude/frequency is adjusted adequately, optical stimulation is less adjustable. However, some researchers claim that, by increasing the intensity of visible light, they can go through some tissue, which is correct, with some limitations due to tissue density. If we compare this technology to pacemakers and deep brain stimulation which are successful technologies, brain optical stimulation still requires many validations before bringing it to a commercial level as a brain aid technology. Indeed, the electrical stimulation can be adjusted depending on the growing tissue, which makes electrical-based stimulation more practical. The growing tissue will change the electrode impedance and, consequently, by sampling the electrode impedance, it is possible to adjust the signal intensity as the tissue itself is electrically conductive.

In conclusion, considering today's technologies, the optical sensors/stimulators are extremely sensitive to the biological environment, in contradistinction with the electrical sensors when applied in biomedical research. Electrical stimulations can also be adjusted more easily than optical stimulations. Finally, while some detection techniques such as SERS have been proven to be able to detect even a single molecule, noble metallic-based optical detection is relatively new and the cytotoxicity of the most used materials is currently unknown.

## 4. Conclusions

This work surveyed the most common neurotransmitters and their sensing techniques. These electrochemical messengers in the nervous system are synthesized and stored in presynaptic neurons, and they are released in the synaptic clef to act on postsynaptic neurons as inhibitors or exciters, thus affecting the transmission of information between neurons. More than 100 neurotransmitters have been identified and they are all essential for proper functioning of the nervous system. Given their

vital role, their dysfunction can lead to serious malfunction of the central nervous system including psychiatric disorders such as schizophrenia, epilepsy, depression, Parkinson's and Huntingdon's disease, among others. Hence, for clinical or neuroscience research purposes, neurotransmitters have to be accurately detected and many techniques have been proposed to do so. However, given the low concentration of neurotransmitters and the large number of other biomolecules and minerals mixed with them, the accurate neurotransmitter detection in the brain is still a challenge. Some techniques such as microdialysis and HPLC have been clinically used for decades, but other methods such as optical fiber sensing and colorimetry are emerging with improved detection time, sensitivity and selectivity at lower cost. However, as neurotransmitters are generally locally produced in a specific area of the brain at a specific time, nuclear medicine tomographic imaging techniques are preferable where the kinetics of metabolism of the target molecule is required.

**Author Contributions:** All authors contributed to the overall review design, literature collection and analysis, and the writing of the manuscript. E.B., M.B., and A.M. contributed to the funding acquisition.

**Funding:** This work has been funded by the National Research Council of Canada (NSERC) and the Fonds de recherche du Québec–Nature et technologies (FRQNT)/Quebec Strategic Alliance for Microsystems (ReSMIQ).

**Acknowledgments:** The authors acknowledge the financial support from the National Research Council of Canada (NSERC) through the discovery grant program—the Fonds de recherche du Québec–Nature et technologies (FRQNT)/Quebec Strategic Alliance for Microsystems (ReSMIQ) through its complementary scholarship program. In addition, the authors are indebted to the Natural Sciences and Engineering Research Council of Canada (NSERC), the CHU de Québec Foundation and the Fonds de recherche du Québec – Nature et technologies (FRQNT)/Quebec Strategic Alliance for Microsystems (ReSMIQ) for their financial support. E.B. is a research scholar from the Fonds de recherche du Québec–Santé (FRQS) in partnership with the Antoine Turmel Foundation.

**Conflicts of Interest:** The authors declare no conflict of interest.

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
