# Peer review of "A Review of Neurotransmitters Sensing Methods for Neuro-Engineering Research"

_applsci, doi:10.3390/app9214719_

Round 1

Reviewer 1 Report

This is a review of neurotransmitter detection methods with a description of important neurotransmitters in the introduction to preface the summary of methods. The paper provides a summary of many techniques.

Below are specific comments about the manuscript.

There are several grammatical and spelling errors in the abstract and throughout the manuscript that can be corrected with a thorough grammar and spelling review.

Lines 44-45 should be modified to indicate that Figure 1 shows the locations where the listed neurotrasmitters are found in the highest concentrations. The way the text reads now makes it sound as if they are only found in the labeled regions.

Given that there are more than 100 neurotransmitters, the authors should consider adding text before section 1.1 to explain why they chose to highlight the neurotransmitters in sections 1.1-1.3.

I recommend the authors remove the words “positive symptoms” and “negative symptoms” from lines 130-131. Assigning positive and negative to the symptoms listed is subjective and also sounds a bit odd.

While the introduction has some nice aspects to it, such as a description of key neurotransmitters, the introduction needs to list other similar reviews that have been written about neurotransmitter sensing and how this review is different and/or why it is needed. As one example, the authors could reference Wu et al, ACS Omega 3 (2018) 13267-13274 and highlight what additional information the present review contributes.

The review of sensing techniques lacks recent references that would give the reader a sense of the current state-of-the-art. For example, the average year for the papers referenced in section 2.1 is 1996 and for section 2.2 the average reference year is 2004, although there are subsections with some more recent papers references such as sections 2.2.6. Using older papers is good to provide a historical perspective but a review paper must focus on the most recent advances or the manuscript will read like a textbook chapter.

This comment should be addressed through a rewriting of section 2 to include references and descriptions of very recent papers (last 5 years) of the most important advances for each sensing method described in Section 2.

Figure 2 is too difficult to read and is also too far away from where it is mentioned in section 2.2.5.

Lines 426-427: I would remove the words “less common” when referring to amperometry compared to FSCV. Amperometry is very often used for neurotransmitter detection and actually predates FSCV.

Lines 446-447: I believe FSCV scan rate can be 400 V/s but is often much higher. Also, cycle duration can be 100 ms but can vary. I recommend describing the method is more general terms.

Line 463: do the authors mean 1400 V/s for scan rate?

Section 2.6.2 needs more information added to it.

While I like the idea of the flow chart in Figure 8, I recommend removing it because there is much overlap between techniques regarding what technique is the best for each criteria and furthermore, the criteria for choosing a detection method are usually much more complicated than is presented in the figure.

Lines 572-573: this statement is not true. Enzymatic modification has been shown in many instances to make detection of neurotransmitters that are not easily electrolyzed.

Lines 584-585: I believe the authors are referring to abundance rather than importance when referring to blood and cerebrospinal fluid.

Lines 588-590: this statement is not quite true. There are commercial blood glucometers that continuously monitor blood glucose using a semi-implanted electrode under the skin for up to 14 days at a time.

Lines 641-643: I do not find this conclusion to be adequately supported by the paper nor is it clear what is clear what is meant by “functional analysis”. It therefore seems inappropriate to end the paper with this statement.

Author Response

We would like to thank you for your time and effort in carefully reviewing our paper, and many helpful suggestions. Our answers are provided in the attached document.

Reviewer 2 Report

The paper entitled "A review of neurotransmitters sensing methods for neuro-engineering research" by Niyonambaza et al. describes techniques for in vitro and in vivo detection of neurotransmitters.

The problematic is clear and the review well written however I'd like to evidence two main points that I think would improve the significance of the review:

- I would suggest an upgrading of the references reported for several techniques described, for example why not to consider in describing nuclear medicine tomographic imaging techniques the paper by Shariatgorji et al published in Neuron in 2014, or in describing fluorescence techniques the paper of Dey et al published in Chemistry in 2017?

- I should be indicated more clearly along the review the kind of specimen for the application of the technique described

Other points:

Regarding the role exerted by glycine it should be mentioned that glycine is a required co-agonist along with glutamate for NMDA receptors.

Minor points:

Lane 45: correct with their

Lane 66: delete role

Line 111: correct with protoplasmic

Line 242: correct with similar

Line 419: delete glycine,

Line 483: correct released

Author Response

(The authors gave the same response as above.)

Reviewer 3 Report

The review article by Miled et al. deals with neurotransmitters and their detection. This topic is very relevant for the life sciences but also fields such as chemistry and physics that contribute to methods development. 

The article is well organised and starts with an overview on neurotransmitters and their function/relevance.Then the different methods available are introduced. Finally, differences and advantages/disadvantages are discussed. 

I think the article adds a novel aspect to the field because it does not focus on one method or neurotransmitter. However, I have several major concerns that should be addressed before the paper can be accepted. 

1) What is the difference to existing review articles ? Please clarify. 

2) A summary as in table 2 is important but should be complemented by numbers. What is the spatial resolution of those approaches and the temporal resolution ? This is essential to assess what a method can resolve. 

3) There are many novel papers missing especially about optical sensing. For example, recently carbon nanotube based nIR fluorescent sensors have been developed and used (in arrays and alone) to detect release  of dopamine. If the authors add such new concepts that could also answer question 1. 

4) Related to question 2. Every method has its merits. Some are very useful to detect static concentrations. Some are more useful to detect transient signals as release from neurons. What is typically missing in most reviews is a perspective on the kinetics of the method and the dynamics of the biological process. It would improve the article if those aspects would be discussed. Maybe also in the context of table 2. I am thinking of papers such as this one  https://pubs.acs.org/doi/10.1021/acsnano.7b00569

5) There has recently been a very impressive paper in Science about aptamer based detection of neurotransmitters. It could be discussed to add another /novel recognition strategy.

Minor: 

Please check the language again. There are many typos. 

Author Response

(The authors gave the same response as above.)

Round 2

Reviewer 1 Report

Thank you for considering and addressing my comments.

Reviewer 2 Report

In the revised version the manuscript has been significantly improved and now warrants publication in Sensors.

Reviewer 3 Report

The authors addressed most of my concerns and is now suitable for publication.